# Comparison of Structure and Properties of Mo_2_FeB_2_-Based Cermets Prepared by Welding Metallurgy and Vacuum Sintering

**DOI:** 10.3390/ma14010046

**Published:** 2020-12-24

**Authors:** Hu Xu, Junsheng Sun, Jun Jin, Jijun Song, Chi Wang

**Affiliations:** Key Laboratory for Liquid-Solid Structural Evolution and Processing of Materials, Ministry of Education, Shandong University, Jinan 250061, China; xh0714@mail.sdu.edu.cn (H.X.); 201733689@mail.sdu.edu.cn (J.J.); 200799013860@mail.sdu.edu.cn (J.S.); Q1817209090@163.com (C.W.)

**Keywords:** VM-Mo_2_FeB_2_, WM-Mo_2_FeB_2_, phase evolution, wear resistance, corrosion behavior

## Abstract

At present, most Mo_2_FeB_2_-based cermets are prepared by vacuum sintering. However, vacuum sintering is only suitable for ordinary cylinder and cuboid workpieces, and it is difficult to apply to large curved surface and large size workpieces. Therefore, in order to improve the flexibility of preparing Mo_2_FeB_2_ cermet, a flux cored wire with 70% filling rate, 304 stainless steel, 60 wt% Mo powder and 40 wt% FeB powder was prepared. Mo_2_FeB_2_ cermet was prepared by an arc cladding welding metallurgy method with flux cored wire. In this paper, the microstructure, phase evolution, hardness, wear resistance and corrosion resistance of Mo_2_FeB_2_ cermets prepared by the vacuum sintering (VM-Mo_2_FeB_2_) and arc cladding welding metallurgy method (WM-Mo_2_FeB_2_) were systematically studied. The results show that VM-Mo_2_FeB_2_ is composed of Mo_2_FeB_2_ and *γ*-CrFeNi.WM-Mo_2_FeB_2_ is composed of Mo_2_FeB_2_, NiCrFe, MoCrFe and Cr_2_B_3_. The volume fraction of hard phase in WM-Mo_2_FeB_2_ is lower than that of VM-Mo_2_FeB_2_, and its hardness and corrosion resistance are also slightly lower than that of VM-Mo_2_FeB_2_, but there are obvious pores in the microstructure of VM-Mo_2_FeB_2_, which affects its properties. The results show that WM-Mo_2_FeB_2_ has good diffusion and metallurgical bonding with the matrix and has no obvious pores. The microstructure is compact and the wear resistance is better than that of VM-Mo_2_FeB_2_.

## 1. Introduction

With its high hardness, high melting point, good wear resistance and corrosion resistance, metal borides are often widely used in mechanical processing, ore grinding, alloy smelting, parts manufacturing and other fields [1,2,3,4]. In the past few years, remarkable achievements have been made in the study of the structures and properties of different types of metal borides, such as the top-down nanostructures of Strontium Hexaboride [5], the phase stability in ternary (Ba_x_Ca_1−x_)B_6_ and (Ba_x_Sr_1−x_)B_6_ compounds [6], and the anisotropic thermal expansion properties of MoAlB, Cr_2_AlB_2_, Mn_2_AlB_2_ and Fe_2_AlB_2_ of selective layered ternary transition metal borides [7]. These ternary boride-based technical ceramics have great potential to replace cemented carbide materials. In particular, Mo_2_FeB_2_ based cermets show superior mechanical properties and economic benefits than other cermets, and have attracted extensive attention compared with other cermets [8,9,10].

The Mo_2_FeB_2_ hard phase is formed by in situ boriding reaction without separate preparation using vacuum liquid phase sintering technology [11,12], which overcomes the disadvantages of poor sinterability and difficultly reacts with binder metal to form brittle phase compared with traditional sintering process. Yu et al. [13] fabricated Mo_2_FeB_2_ based cermets using vacuum sintering at different temperatures with different holding time and revealed that the increase of sintering temperature or holding time promoted the development of elongated Mo_2_FeB_2_ grains when the sintering temperature was below 1250 °C. In addition, further studies showed that the contents of alloying elements significantly decreased the grain size [14,15]. Li et al. [16] added WC in Mo-Fe-B-Cr-Ni system carrying out the vacuum sintering to form the Mo_2_FeB_2_-WC double hard phases, which improved the bending strength and hardness of the material.

At present, the vacuum liquid phase sintering method is mostly used to prepare Mo_2_FeB_2_ based ceramics, but this method is only limited to ordinary cylinders and cuboids, and the flexibility is not enough, so it is relatively difficult to prepare large curved surface and large-size workpieces. Argon arc welding has achieved good results in the preparation of such workpieces [17,18,19]. Therefore, in order to improve its flexibility, flux cored wire can be prepared and Mo_2_FeB_2_ cermet can be prepared on the surface of workpiece by argon arc welding metallurgy. This method can be applied to any curved surface of any shape and can also produce a large area of workpiece surface, which greatly improves its application range.

## 2. Materials and Methods

The material used for preparing Mo_2_FeB_2_-based cermets by an arc cladding welding metallurgical method (WM-Mo_2_FeB_2_) is flux-cored wire, and the technical parameters of alloy powder used for preparing Mo_2_FeB_2_ cermet by vacuum sintering (VM-Mo_2_FeB_2_) are shown in Table 1.

### 2.1. WM-Mo2FeB2 Preparation

The diameter of the flux-cored wire designed in this study is 3.2 mm, the powder composition of the flux-cored wire is 60 wt% Mo and 40 wt% FeB, and the powder filling rate is 70%. The steel strip of flux-cored wire is 304 stainless steel. The technical parameters of the alloy powder are shown in Table 1, and composition of the 304 stainless steel is shown in Table 2.

The preparation process of flux cored wire is as follows:Weigh the alloy powder according to the mass ratio of 60 wt% Mo and 40 wt% FeB and then mixed for 10 min to obtain the powder;After 304 stainless steel is cleaned by ultrasonic cleaning equipment, the steel strip is rolled into U-shape by flux cored wire production equipment, and the powder made in step (1) is added into the U-shaped steel strip;The U-shaped groove is closed to make the powder wrapped in it, and the closure part is connected by lap joint; Through wire drawing, drawing and reducing are carried out step by step, and finally the diameter reaches 3.2 mm;The flux cored wire layer obtained in step (3) is wound into a coil to obtain the finished surfacing flux cored wire.

After the flux cored wire is prepared, it is cladding on the Q235 substrate with the arc of the ZX7-315STG argon arc welding machine to prepare WM-Mo_2_FeB_2_. In the welding process (TIG), pure argon is mainly used as shielding gas, and tungsten electrode moves at the speed of 100 mm/min. After the sample is cooled to room temperature, the second layer and the third layer are deposited on the first layer with the same process parameters to ensure that the cladding layer has a certain thickness, thus reducing the influence of base metal on the dilution of cladding metal. See Table 3 for welding parameters.

### 2.2. VM-Mo2FeB2 Preparation

Commercial Mo powders, FeB powders, Ni powders, Cr powders and carbonyl Fe powders were used as raw materials. Table 4 shows the chemical composition and particle size of the alloy powders.

The composition design (mass fraction) of the alloy powders was 42% Mo+28% FeB+5% Cr+3% Ni + balance Fe. Alloy powders were mixed in QM-3SP2 (Shanghai Shupei Experimental Equipment Co.,Ltd., Shanghai, China) planetary ball mill for 24 h at a speed of 600 r/min. Polyvinyl Butyral (PVB) (Jinzhou Hongda New material Co.,Ltd., Jinzhou, China), which accounted for 3% of the total mass of the alloy powders, was used as binder. The mixture was pressed into pressure machine with dimensions of 400 mm × 40 mm × 4 mm at a pressure of 150 MPa. After drying, it is sintered in a vacuum sintering furnace and the degree of sintering vacuum was 1.0×10^−2^ ~ 1.0×10^−3^ Pa. The sintering system is: Increasing the temperature from room temperature to 400 °C at a speed of 10 °C/min, after 30 min incubation, increasing the temperature up to 1050 °C at a speed of 10 °C/min for 30 min, the temperature up to 1210 °C at a speed of 5 °C/min for 30 min, and then cooled to room temperature with the furnace to obtain VM-Mo_2_FeB_2_.

### 2.3. Microstructural Investigations and XRD Analysis

To observe its microstructure, the WM-Mo_2_FeB_2_ and VM-Mo_2_FeB_2_ were ground with 1500 mesh continuous grade sandpaper and polished by 1.5 μm diamond paste. Then, the specimens were etched using a 20 vol% hydrofluoric acid, 30 vol% hydrochloric acid and 50 vol% nitric acid solution for 10 s. The microstructure was studied with scanning electron microscope (SEM, JSM-6600V, Japanese electronics company, Tokyo, Japan.) in backscattered electron (BSE) mode while analysis of the chemical composition of both hard phase and binder phase was performed by an electron probe micro analyzer (EPMA, JXA-8530F PLUS, Japanese electronics company, Tokyo, Japan). The volume fraction of microstructure constituents was calculated by the Axioimaging software related to Axio Lab A1 optical microscope. X-ray diffraction (XRD, Jinan, China) was used to analyze the phases of the WM-Mo_2_FeB_2_ and VM-Mo_2_FeB_2_.

### 2.4. Hardness Measurement

The microhardness of hard phases in the WM-Mo_2_FeB_2_ and VM-Mo_2_FeB_2_ were texted by a Vickers hardness tester with a load of 0.5 kg for 10 s dwell time. The rockwell hardness of the WM-Mo_2_FeB_2_ and VM-Mo_2_FeB_2_ were also measured. At least three points in samples were measured randomly and the average value was set as the final result.

### 2.5. Wear Tests

For the wear resistance examination, the samples were machined to 31 mm × 7 mm × 5 mm and smoothed the sample surface with abrasive papers to remove metallic oxide of surface. Block on ring wear resistance tests were performed using an MM200 testing machine (Hebei Xuanhua Zhengli Balancing Machine Co.,Ltd., Hebei, China) under dry and rotating condition at room temperature. Carburized 20CrMnTi steel with the size of 40 mm in diameter and 10 mm in thickness was selected as the counterpart, whose Rockwell hardness was 60 HRC (Rockwell hardness). The wear test speed was 200 r/min, the load was 150 N and the wear time was 60 min, respectively. The worn morphologies of the samples were analyzed using SEM to determine the wear mechanism.

### 2.6. Corrosion Resistance Test

Corrosion resistance of the samples was investigated through potentiodynamic polarization on CS350 electrochemical workstation. In this study, a conventional three-electrode system with platinum plate, saturated calomel electrode (SCE) and samples acted as counter electrode, reference electrode and working electrode, respectively. Potentiodynamic polarization tests were implemented in 3.5 wt% NaCl solution at room temperature with a scanning rate of 0.5 mV/s. The scanning range was −0.65 to 0.25 V relative to the open circuit potential (OCP) and at least three sets of measurements of samples were carried out to ensure the repeatability. Moreover, the corrosion morphologies of the WM-Mo_2_FeB_2_ and VM-Mo_2_FeB_2_ after the electrochemical tests were investigated by SEM and the composition of the corrosion products produced on the samples was investigated through X-ray photoelectron spectroscopy (XPS, Jinan, China). Photoelectron emission was excited by monochromatic Al K_α_ (1486.6 eV) source. The C 1 s peak from adventitious carbon at 284.8 eV was used as a reference to correct the charging shifts.

## 3. Results and Discussion

### 3.1. XRD Analysis

The XRD patterns of the WM-Mo_2_FeB_2_ and VM-Mo_2_FeB_2_ are shown in Figure 1.

The VM-Mo_2_FeB_2_ is mainly composed of *γ*-CrFeNi and Mo_2_FeB_2_ (Figure 1a). No other new phases were formed during the sintering process of ternary borides. At the same time, comparing the diffraction peak intensity of the two main phases, it is found that the diffraction peak intensity of *γ*-CrFeNi phase is weaker than that of Mo_2_FeB_2_ phase, indicating that the content of Mo_2_FeB_2_ phase in VM-Mo_2_FeB_2_ is relatively higher than that of the *γ*-CeFeNi phase. In addition, no brittle diffraction was detected in Fe_2_B. Because of its poor crystallinity [20], the large area segregation of Fe_2_B brittle phase will destroy the original integrity and continuity of Mo_2_FeB_2_-based cermets. Therefore, when subjected to load, cracks will occur and rapidly expand, thus affecting the mechanical properties of the material [21]. Figure 1b shows the XRD diffraction pattern of WM-Mo_2_FeB_2_. The results show that the WM-Mo_2_FeB_2_ is composed of Mo_2_FeB_2_ (M_3_B_2_), NiCrFe, MoCrFe and Cr_2_B_3_. Mo_2_FeB_2_ forms M_3_B_2_ compound boride with Mo and Cr at high temperature. At the same time, MoCrFe is a solid solution of Mo and Cr dissolved in Fe, while NiCrFe is a solid solution formed by solid solution of Ni and Cr in Fe. Besides the main phase, Cr_2_B_3_ boride was also found in the WM-Mo_2_FeB_2_. The XRD patterns of the WM-Mo_2_FeB_2_ and VM-Mo_2_FeB_2_ were compared, and there were some differences between them. On the one hand, the number of phases in the WM-Mo_2_FeB_2_ increased rapidly with the appearance of MoCrFe phase and Cr_2_B_3_ phase. On the other hand, from the diffraction peak intensity, the content of the Mo_2_FeB_2_ hard phase formed in the WM-Mo_2_FeB_2_ is obviously lower than that of VM-Mo_2_FeB_2_.

In order to further confirm the Mo_2_FeB_2_ phase and its main form and distribution in the VM-Mo_2_FeB_2_, an electron probe micro analyzer (EPMA) was used to analyze the composition of VM-Mo_2_FeB_2_. From the BSE image of VM-Mo_2_FeB_2_ (Figure 2), it consisted mainly of a white phase and we performed component analysis on the white phase (one zone in Figure 2); the results are listed in Table 4.

It was found that the white phase was mainly composed of Mo, Fe, B, and Cr and the atomic ratio of Mo, Fe and B did not satisfy 2:1:2. This was because chromium substitutes for Mo and Fe sites in the lattice that form complex borides M_3_B_2_(M: Mo Fe Cr) according to the reactions [22,23] and the Equations (1) and (2) are as follows:Cr + Mo_2_FeB_2_ → (Mo, Fe, Cr)_3_B_2_(1)
Fe + Fe_2_B + (Mo, Fe, Cr)_3_B_2_ → Liquid + (Mo, Fe, Cr)_3_B_2_(2)

According to the ternary phase diagram of Mo-Fe–B [22], the liquid phase is Fe-B eutectic liquid phase.

Mo_2_FeB_2_ belongs to the tetragonal lattice [24] and its crystal structure is shown in Figure 3.

The lattice constants of Mo_2_FeB_2_ are a = b = 5.807 Å, C = 3.142 Å. In Mo_2_FeB_2_ protocell, there are eight Fe atoms at the vertex position, two at the face center position, four Mo atoms and all located in the cell, and eight B atoms are all on the surface [25].

After analysis, the atomic ratio of M (M: Mo, Fe, Cr) to B is close to 3:2, combined with the XRD pattern, thereby we presume that the white phase is complex ternary boride M_3_B_2_.

### 3.2. Microstructure Characterization

The BSE image of VM-Mo_2_FeB_2_ is shown in Figure 4a.

It can be seen from the figure that most of the VM-Mo_2_FeB_2_ is equiaxed, and the volume percentage is significantly higher than that of the bonded phase *γ*-CrFeNi. The hard phase particles are smaller and the growth is incomplete, which is due to the lower solid-state sintering temperature in the process of solid-phase sintering. Yu et al. [26] revealed that Mo_2_FeB_2_-based cermets will evolve from equiaxed grains of solid phase sintering to flake or columnar shape of liquid phase sintering during sintering. At the same time, it can be seen from the image that there are a lot of pores in the VM-Mo_2_FeB_2_, and the density is relatively low. This is because the sintering time is very short, and the liquid phase generated by the eutectic reaction does not have enough diffusion to fill these pores during the solid-liquid transition. Figure 4b–d show the typical microstructure of different areas of WM-Mo_2_FeB_2_. It can be seen from the figure that the WM-Mo_2_FeB_2_ is mainly composed of various massive, fishbone-shaped and irregular hard phases and eutectic matrix. The hard phase on the surface of the WM-Mo_2_FeB_2_ is distributed in the eutectic matrix as coarse blocks (Figure 4b). The coarsening of massive structure can be used to explain this phenomenon, that is the dissolution cycle of hard phase of fine particles in VM-Mo_2_FeB_2_ under the action of arc heat, and then through liquid phase diffusion; small hard particles precipitate on the surface of large particles, resulting in the increase of hard phase particle size, which is consistent with the dissolution precipitation mechanism [27]. In the middle layer of the WM-Mo_2_FeB_2_, in addition to the thick block hard phase, there are also fishbone-like hard phase connected by slender hard phase. This is mainly due to the inhomogeneous composition of the liquid phase of the alloying elements Mo, Fe, Cr, B in the front of hard phase interface. At the same time, the (100) direction of Mo_2_FeB_2_ crystal, that is, the c-axis direction, is the preferred direction for grain growth [28]. The above two factors cause the Mo_2_FeB_2_ phase to present a slender shape and connect to form fishbone shape, as shown in Figure 4c. In addition, there is an obvious transition layer near the fusion line in Figure 4d, which indicates good diffusion and metallurgical bonding between the WM-Mo_2_FeB_2_ and the substrate. Compared with VM-Mo_2_FeB_2_, the amount of the hard phase in WM-Mo_2_FeB_2_ is less, and the hard phase is coarser, but there is no obvious porosity, and the microstructure is more dense.

To better analyze the alloying element distribution of the WM-Mo_2_FeB_2_, the map scanning images of the WM-Mo_2_FeB_2_ with different areas are shown in Figure 5.

As we can see from the figure, elemental Mo and B are mainly distributed in the hard phase while elemental Fe is mainly distributed in the matrix. The distribution of Cr and Ni in different regions of the WM-Mo_2_FeB_2_ is also different, for the top and middle layers of the WM-Mo_2_FeB_2_, elemental Cr and Ni are mainly distributed in the matrix. While at the bottom of the WM-Mo_2_FeB_2_, elemental Cr and Ni are uniformly distributed but there is no behavior of segregation.

Figure 6 shows elemental content changes along the longitudinal direction of the WM-Mo_2_FeB_2_, the content of Mo decreases sharply from the top layer to the bottom layer of the WM-Mo_2_FeB_2_ while the counterpart of Fe element increases dramatically.

Some reasons can explain this phenomenon: The welding current used in WM-Mo_2_FeB_2_ was large (140A) and the heat input was large, which can quickly melt the surface of the carbide-tipped electrode and substrate to form a weld pool. Under the agitation of the arc, the molten liquid substrate and the weld pool of the bottom region mixed so that the concentration of Mo in the bottom layer was greatly reduced while Fe greatly increased. Meanwhile, the flow rate of the liquid metal in the bottom area was relatively slow and temperature was low while the viscosity was large, which made the molten metal of the Mo-rich difficult to flowing to the bottom of the weld pool near the fusion line so that the Mo content was low while the Fe content was higher resulting in a poor Mo rich Fe atmosphere. While the B element slowly increased, possibly due to the lighter mass of B element and small buoyancy while the cooling rate of the welding was faster, some B elements were less than floating and finally remained at the bottom. The deeper the Cr element reached the bottom of the weld pool and the slight decrease in elemental content may be attributed to the dilution of the matrix and the change in Ni element content was relatively not obvious. The concentration of Mo in the top layer was large and Mo_2_FeB_2_ easily formed. As the depth increased, Mo_2_FeB_2_ gradually decreased, which was basically consistent with the microstructure analysis (Figure 4).

### 3.3. Phase Evolution in Sintering and Welding

#### 3.3.1. Sintering Phase Evolution

Previous literature studies [29,30] indicated that Mo_2_FeB_2_ formed using solid phase in situ reaction between powder particles in the ternary boride solid phase sintering process, and the phase transformations involved in the process are listed as follows, Equations (3)–(5):FeB + Fe = Fe_2_B(3)
2Mo + 2FeB = Mo_2_FeB_2_+ Fe(4)
2Mo + 2Fe_2_B = Mo_2_FeB_2_+ 3Fe(5)

After milling, the binary boride FeB and the metal powder were thoroughly mixed, as shown in Figure 7a.

When the temperature arrived 452 °C, the transformation from FeB to Fe_2_B occurs (Equation (3)) [30], as shown in Figure 7b. While at 852 °C, the transformation form binary borides to Mo_2_FeB_2_ happens in Mo enrichment atmosphere (Equations (4) and (5)), as shown in Figure 7c and the Mo_2_FeB_2_ hard phases precede the liquid phase formation. At the same time, as the temperature increases further, the diffusion of Mo, B, Fe and other elements accelerates, which makes the Mo_2_FeB_2_ hard phases grow (Figure 7d). Subsequently to the process of liquid phase sintering, the following transformations occur, Equations (6) and (7):γ-Fe + Fe_2_B = L_1_(6)
γ-Fe + L_1_ + Mo_2_FeB_2_ = L_2_ + Mo_2_FeB(7)

When the temperature reached 1092 °C, the generation of eutectic liquid phase L_1_ (Equation (6)) appeared around Mo_2_FeB_2_ hard phases and dissolved and rearranged Mo_2_FeB_2_ hard phases as well as performed initial densification on the VM-Mo_2_FeB_2_, as shown in Figure 7e. As the temperature increased, Equation (7) applied. Mo_2_FeB_2_ hard phases have a high solubility in the liquid phase L_2_ and further densify under the influence of L_2_. Finally, the VM-Mo_2_FeB_2_ mainly consists of Mo_2_FeB_2_ hard phase and γ-CrFeNi, as shown in Figure 7f.

#### 3.3.2. Welding Phase Evolution

In the process of argon arc cladding, the formation of Mo_2_FeB_2_ cermet conforms to the nucleation and growth mechanism. The first stage is the full mixing of metal powders. In the second stage, the metal powder gradually melts under the action of welding heat, and then forms the welding pool. According to the Fe–B binary phase diagram [31], the formation temperatures of FeB and Fe_2_B are 1650 °C and 1389 °C, respectively. Mo atom in liquid phase reacts with FeB and Fe_2_B to form Mo_2_FeB_2_ (equations 4 and 5), and then precipitates hard phase and is covered by liquid phase surface.Under the synergistic effect of high temperature and alloy element concentration, high concentration Cr element in liquid phase diffuses to Mo_2_FeB_2_ (formula 1) [29]. Cr atoms replace Mo and Fe atoms in Mo_2_FeB_2_ lattice, resulting in lattice distortion and effectively improving the crystal structure of Mo_2_FeB_2_ mechanical properties [32]. With the continuous decrease of temperature, element B is basically involved in the formation of boride, rather than the solid solution in the matrix [33]. When the eutectic temperature is reached, the residual liquid phase will solidify to form Fe-Cr matrix.

### 3.4. Hardness Analysis

Figure 8 shows the Rockwell hardness and Vickers hardness of VM-Mo_2_FeB_2_ and WM-Mo_2_FeB_2_.

In order to make the data accurate, each test data was measured five times, and the average value was considered. At the same time, the standard deviation of the data was analyzed by using the principle of statistics. The final result is shown in Figure 8. The Rockwell hardness of VM-Mo_2_FeB_2_ and WM-Mo_2_FeB_2_ is 62 HRC and 60.5 HRC respectively. The Vickers hardness is 842.4 HV_0.5_ and 828.9 HV_0.5_, respectively. Although the hardness of the WM-Mo_2_FeB_2_ decreases, the hardness difference between the VM-Mo_2_FeB_2_ and WM-Mo_2_FeB_2_ is not obvious, which is mainly due to the increase of the hardness of the matrix by the solution strengthening of chromium, nickel and other alloy elements in the welding process. In addition, the formation of hard phase Mo_2_FeB_2_ also improves the hardness of WM-Mo_2_FeB_2_. Compared with VM-Mo_2_FeB_2_, the hardness of WM-Mo_2_FeB_2_ decreases slightly, which is mainly related to the content of hard phase Mo_2_FeB_2_. The volume fraction of hard phase was calculated for each sample. Red indicates the matrix phase, and the other color belongs to Mo_2_FeB_2_ hard phase (Figure 9).

In Figure 9a, the proportion of Mo_2_FeB_2_ hard phase is 76.13%, while in Figure 9b, the corresponding proportion is only 69.05%. The amount of hard phase in VM-Mo_2_FeB_2_ is much more than that in WM-Mo_2_FeB_2_. Combined with the previous microstructure analysis (Figure 4), it can be found that the number of hard phases in the WM-Mo_2_FeB_2_ is significantly reduced, and some hard phases in the VM-Mo_2_FeB_2_ are connected with each other, which is the reason why the hardness of VM-Mo_2_FeB_2_ is higher than that of WM-Mo_2_FeB_2_.

### 3.5. Wear Properties

#### 3.5.1. Wear Measurement

The weight loss and wear time curve of VM-Mo_2_FeB_2_ and WM-Mo_2_FeB_2_ are shown in Figure 10.

It can be seen that in the first 20 min of the wear test, the weight loss of both is relatively large. This is because it is difficult to achieve the desired smoothness between the contact surface of the specimen and the wearing material. There are randomly distributed micro bumps of different sizes, but the actual contact phenomenon only exists between the micro bumps, and then the stress amplitude modulation phenomenon [34] will appear, resulting in the actual contact area less than the nominal contact area, and each rough body bears more load. According to the mechanical locking theory proposed by Amontons, the dynamic friction is caused by the upper micro convex body passing through the lower micro convex body, which eventually causes the micro convex body to fall off from the material surface and further increase the wear amount. With the wear going on, the asperities gradually wear and the actual contact area increases. After 30 min of wear test, the wear amount enters the stable wear stage, and the wear amount gradually decreases in unit time, and the final wear amount tends to be stable. The wear loss of VM-Mo_2_FeB_2_ is 10.5 mg, and the WM-Mo_2_FeB_2_ is 2.4 mg. Its wear resistance is better than that of VM-Mo_2_FeB_2_. Generally speaking, the wear resistance increases with the increase of hardness and volume fraction of carbide or boride. The hardness and volume fraction of VM-Mo_2_FeB_2_ are higher than that of WM-Mo_2_FeB_2_. In fact, during the wear process of VM-Mo_2_FeB_2_, the bonding force between the hard phase particles protruding from the surface and the matrix is less than the shear stress of the worn surface. Some hard phases fall off from the matrix to form new abrasives, which aggravates the abrasive wear. The results show that the wear resistance is not only related to the hardness and volume fraction of the hard phase, but also to the microstructure and distribution of the hard phase.

#### 3.5.2. Wear Morphologies and Wear Mechanism

Figure 11 shows the wear morphology of VM-Mo_2_FeB_2_ and WM-Mo_2_FeB_2_.

In Figure 11a, it can be clearly seen that there are obvious wear grooves around Mo_2_FeB_2_, which is the characteristic of abrasive wear. Moreover, wear traces of VM-Mo_2_FeB_2_ can be found around Mo_2_FeB_2_, which also shows that the wear performance of VM-Mo_2_FeB_2_ is inferior to that of WM-Mo_2_FeB_2_. In Figure 11b, there is a large amount of wear debris. This is mainly due to the slight deformation of the matrix material under the action of extrusion and shear. Partial deformation causes the atoms in the contact area to approach each other and produce partial adhesion. When the sliding continues, the attachment points are sheared and transferred to the surface of the substrate and fall off to form wear debris, which is a typical feature of adhesive wear.

### 3.6. Corrosion Resistance Analysis

#### 3.6.1. Polarization Curves

The corrosion behavior of VM-Mo_2_FeB_2_ and WM-Mo_2_FeB_2_ in 3.5 wt% NaCl solution was studied by potentiodynamic polarization technique. The polarization curve is shown in Figure 12.

Obvious passivation phenomenon is observed in VM-Mo_2_FeB_2_ and WM-Mo_2_FeB_2_, and the passivation range is about 0.2 V, indicating the formation of stable passivation film. When the potential increases to about 0 V, the current density increases sharply, which indicates that the passive film breaks and the corrosion behavior continues. Table 5 shows the corrosion parameters obtained from potentiodynamic polarization test.

From the corrosion potential measurements, the corrosion potential of VM-Mo_2_FeB_2_ is slightly higher than that of WM-Mo_2_FeB_2_, which indicates that the corrosion tendency of WM-Mo_2_FeB_2_ is higher than that of VM-Mo_2_FeB_2_. From the view of corrosion current density, the corrosion current density of VM-Mo_2_FeB_2_ is much lower than that of WM-Mo_2_FeB_2_, which indicates that WM-Mo_2_FeB_2_ has fast corrosion rate and poor corrosion resistance. Pitting potential is the critical potential which causes the corrosion of metal electrode and the sharp increase of corrosion current density. The pitting potential generally reflects the stability of the passive film. Generally speaking, the higher the pitting potential, the better the stability of the passive film. It can be seen from the results in Table 5 that the pitting potential of VM-Mo_2_FeB_2_ is higher than that of WM-Mo_2_FeB_2_, indicating that the passive film of VM-Mo_2_FeB_2_ is more stable and has better corrosion resistance.

#### 3.6.2. XPS Analysis

The chemical composition of the passive film was studied by XPS. Figure 13 shows the detailed spectra of Fe2p, O1s and Cr2p in the passive film formed on VM-Mo_2_FeB_2_ and WM-Mo_2_FeB_2_ in 3.5 wt% NaCl solution.

As we can see, the passive films of VM-Mo_2_FeB_2_ and the WM-Mo_2_FeB_2_ both have the same chemical composition and the observed XPS pattern indicates the presence of iron, chromium, oxygen in the passive film. The spectra of Fe2p (Figure 13a) shows the presence of two components: metallic, Fe^0^ (706.6 eV) and Fe_2_O_3_ (710.9 eV) [35,36]. The spectra of O1s (Figure 13b) indicates two components: oxide and hydroxide. The peak at 531.3 eV corresponds to OH^-^ in the hydroxide and a second one 530.2 eV corresponds to O^2-^ in the oxide [35]. In Cr2p spectra (Figure 13c), two components can be observed: Cr^3+^ in Cr_2_O_3_ (576.5 eV) and Cr^3+^ in Cr(OH)_3_ (577.2 eV) [35]. Therefore, the components of the passive film are mainly Fe_2_O_3_, Cr_2_O_3_ and Cr(OH)_3_.

#### 3.6.3. Corrosion Morphologies

Corrosion morphologies of VM-Mo_2_FeB_2_ and WM-Mo_2_FeB_2_ after potentiodynamic polarization test are shown in Figure 14. As can be seen from the figure, the corrosion is likely to occur at the boundaries between the Mo_2_FeB_2_ phase and the iron matrix and tends to the matrix with less Mo content. This is because the Mo_2_FeB_2_ hard phases itself have good corrosion resistance [37], which reduce the corrosion of chloride ions in the solution. Moreover, the corrosion resistance of a certain material is usually closely related to the chemical composition. As is well known, chemical elements such as Cr and Mo are the crucial factors to improve the pitting corrosion resistance of stainless steel. They do not only reduce the generation capacity of pitting nuclear but also decease the growth rate of pitting pits [38]. Compared with hard phases, the less solid solution Cr and Mo elements in the matrix and the uneven distribution of element components caused by liquid weld pool, which lead to corrosion occurs on the matrix where the content of Cr and Mo is small. In addition, compared with VM-Mo_2_FeB_2_, the size of the corrosion pits of the WM-Mo_2_FeB_2_ is relatively larger. This is because the liquid weld pool has a short duration and insufficient element diffusion during the welding process, which make the element distribution uneven resulting in poor corrosion resistance of the WM-Mo_2_FeB_2_ in the areas with poor Cr and Mo elements. XPS analysis (Figure 13) shows that the passive film (Cr_2_O_3_, etc.) on the surface of the WM-Mo_2_FeB_2_ in those area is not sufficient to block the penetration of reactive anions (Cl^-^), which penetrate through the passive film to further corrosion and appear pitting spots. Under the continuous action of anode current, reactive anions (Cl^-^) migrate and accumulate at pitting spots and the metal matrix will be deeply etched downward resulting in pitting pits. As the corrosion reaction continues, the dissolved metal ions will increase continuously and under the influence of the downward gravity of the corrosion medium, the pitting pits will expand and develop towards the depth direction.

## 4. Conclusions

In this study, 304 stainless steel strips and appropriate amounts of powder were used to make flux cored wire. WM-Mo_2_FeB_2_ was obtained by argon arc welding on Q235 substrate. VM-Mo_2_FeB_2_ was synthesized in situ by vacuum liquid phase sintering. The microstructure and properties of WM-Mo_2_FeB_2_ and VM-Mo_2_FeB_2_ were systematically studied. The results are summarized as follows:

The VM-Mo_2_FeB_2_ is mainly composed of γ-CrFeNi and Mo_2_FeB_2_, and the WM-Mo_2_FeB_2_ is mainly composed of Mo_2_FeB_2_ (M_3_B_2_), NiCrFe, MoCrFe and Cr_2_B_3_. The hard phase of Mo_2_FeB_2_ in VM-Mo_2_FeB_2_ is equiaxed, while the WM-Mo_2_FeB_2_ is mainly composed of massive, fishbone and irregular Mo_2_FeB_2_ hard phases. The amount of hard phase in VM-Mo_2_FeB_2_ is more than that in WM-Mo_2_FeB_2_.Some pores will be produced in the sintering process of VM-Mo_2_FeB_2_, which affects the final properties. There is a good diffusion and metallurgical bonding between the WM-Mo_2_FeB_2_ and the matrix, and the microstructure is more compact.The hardness of VM-Mo_2_FeB_2_ is higher than that of WM-Mo_2_FeB_2_ due to the higher content of hard phase in VM-Mo_2_FeB_2_, but there is little difference between them. However, the wear resistance is not only affected by the hardness and volume fraction of the hard phase, but also by the morphology and distribution of the hard phase. In the wear test of VM-Mo_2_FeB_2_, some hard phases fall off from the surface of the substrate to form abrasives, which aggravate the wear and make its wear resistance lower than that of WM-Mo_2_FeB_2_.The results show that the corrosion resistance of VM-Mo_2_FeB_2_ is better than that of WM-Mo_2_FeB_2_. The passive films formed on VM-Mo_2_FeB_2_ and WM-Mo_2_FeB_2_ are Fe_2_O_3_, Cr_2_O_3_ and Cr(OH)_3_.

The microstructure and properties of VM-Mo_2_FeB_2_ produced by sintering and the WM-Mo_2_FeB_2_ formed by argon arc welding are systematically studied. It is found that it is feasible to perform cladding on a Q235 substrate by preparing flux cored wire and argon arc welding. In terms of practicability, vacuum sintering can only be used for ordinary cylinders and cuboids, so it is difficult to prepare large curved surface and large size cermets workpieces. Argon arc welding is a good solution to this problem; it is easy to operate and flexible in application. Therefore, through the preparation of flux cored wire, the preparation of cladding layer by welding should be popularized.

## Figures and Tables

**Figure 1 materials-14-00046-f001:**
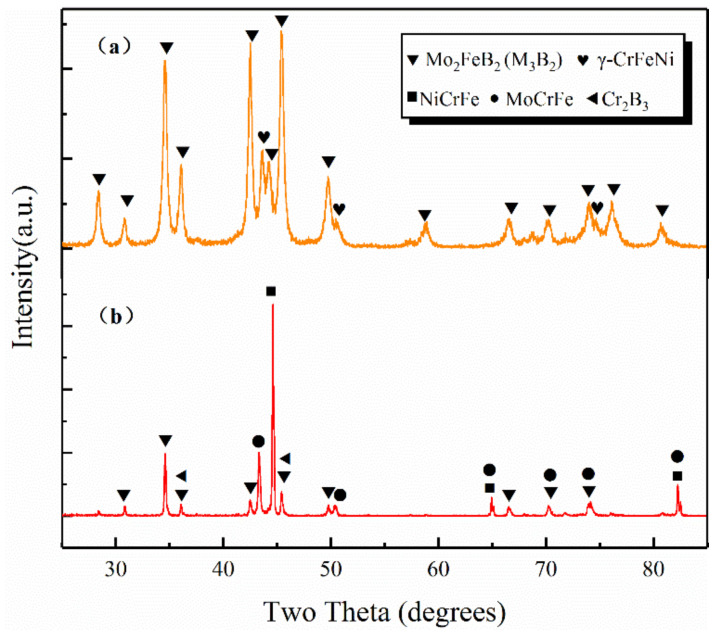
XRD patterns of different samples: (**a**) VM-Mo2FeB2 (**b**) WM-Mo2FeB2 Mo2FeB2(M3B2)(JPCDS-18-0839) γ-CrFeNi(JPCDS-33-0397) NiCrFe(JPCDS-35-1375) MoCrFe(JPCDS-08-0200) Cr2B3(JPCDS-37-1447).

**Figure 2 materials-14-00046-f002:**
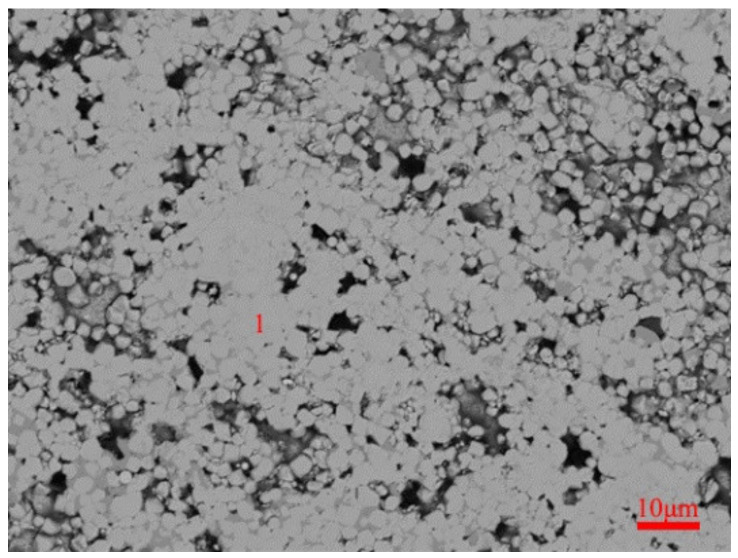
BSE images of VM-Mo_2_FeB_2_.

**Figure 3 materials-14-00046-f003:**
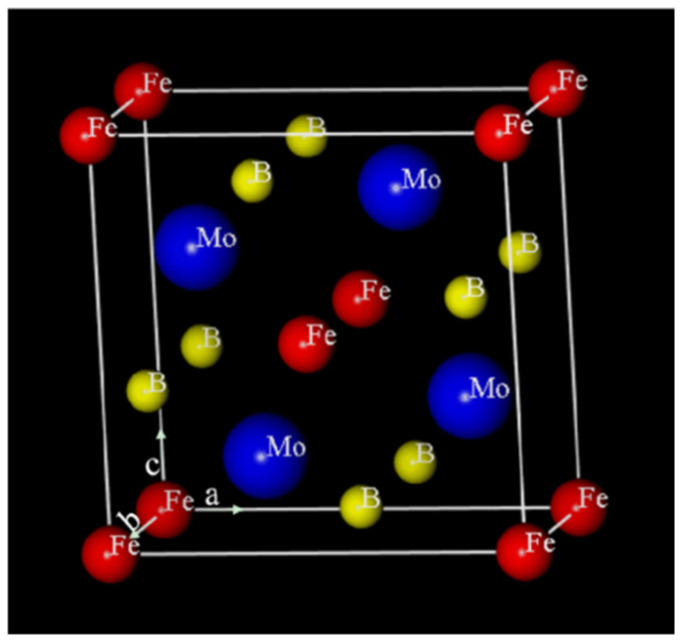
Crystal structure model of Mo_2_FeB_2_.

**Figure 4 materials-14-00046-f004:**
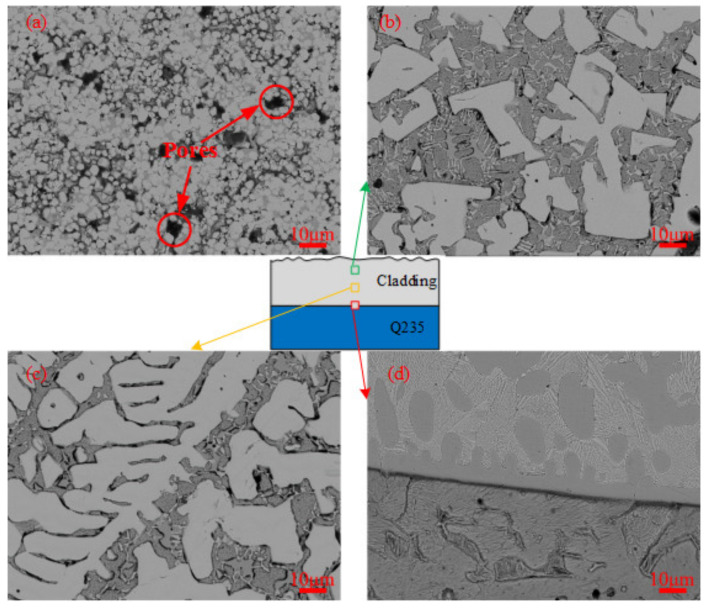
BSE images of different samples: (**a**) VM-Mo_2_FeB_2_, (**b**) the top layer of WM-Mo_2_FeB_2_, (**c**) the middle layer of WM-Mo_2_FeB_2_, (**d**) the fusion zone of WM-Mo_2_FeB_2_.

**Figure 5 materials-14-00046-f005:**
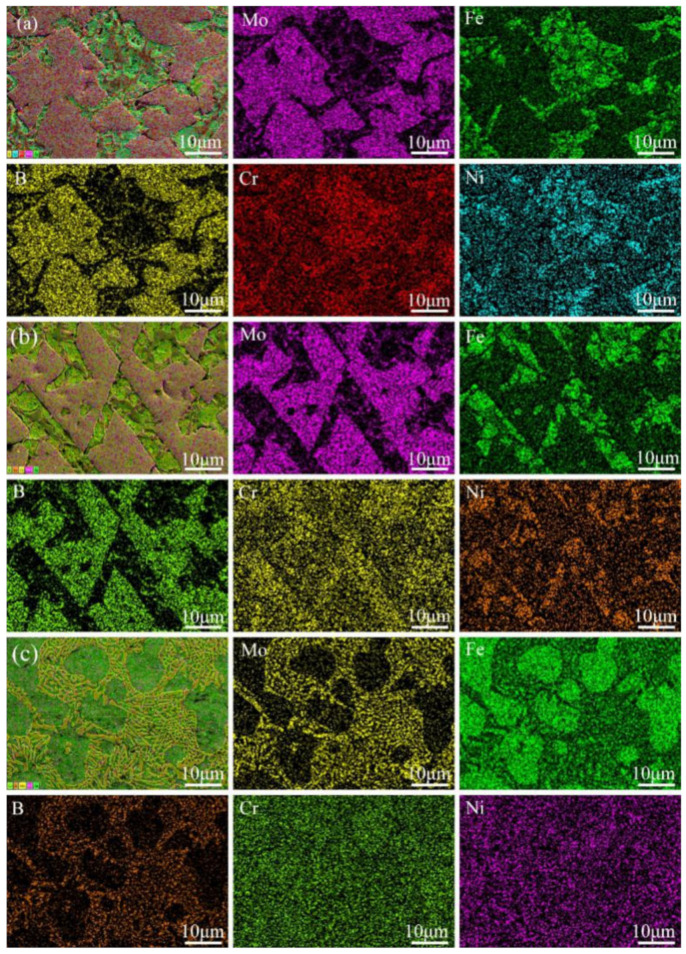
Elements distribution of the WM-Mo_2_FeB_2_ with different zones:(**a**) Top layer, (**b**) Middle layer and (**c**) Bottom layer.

**Figure 6 materials-14-00046-f006:**
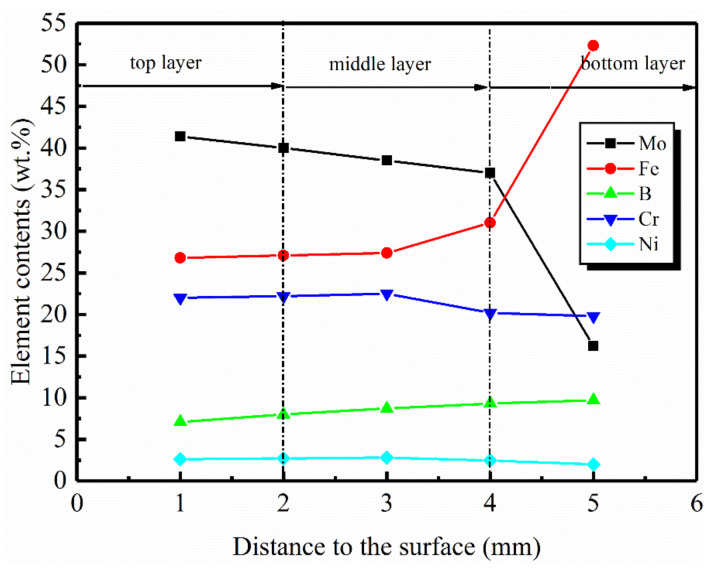
Variation of elemental content of along the longitudinal direction of the WM-Mo_2_FeB_2_.

**Figure 7 materials-14-00046-f007:**
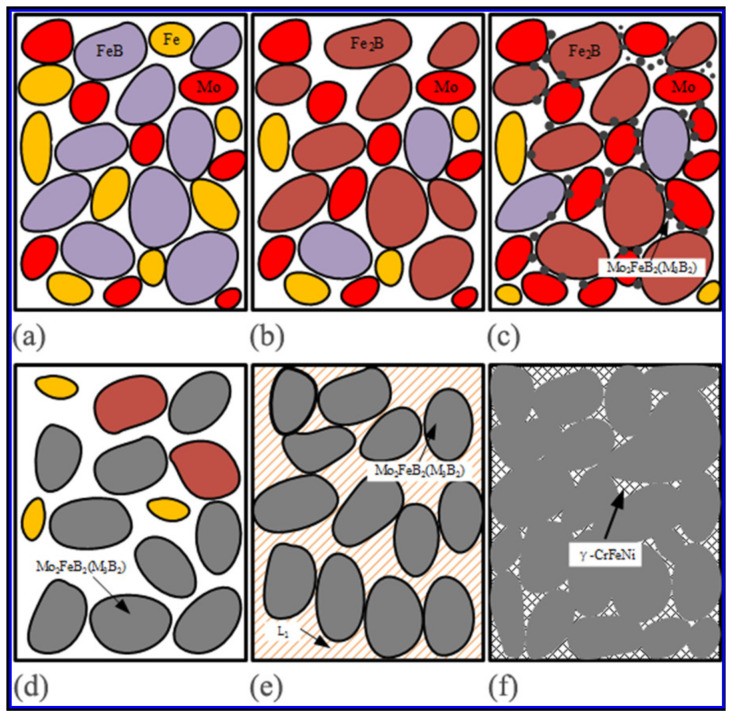
Phase evolution diagram of vacuum liquid phase sintering process (**a**) Powder mixing, (**b**) The transformation of FeB, (**c**) Formation of Mo_2_FeB_2_, (**d**) Growth of Mo_2_FeB_2_, (**e**) Reforming process of Mo_2_FeB_2_ particles, (**f**) Densification of Mo_2_FeB_2_.

**Figure 8 materials-14-00046-f008:**
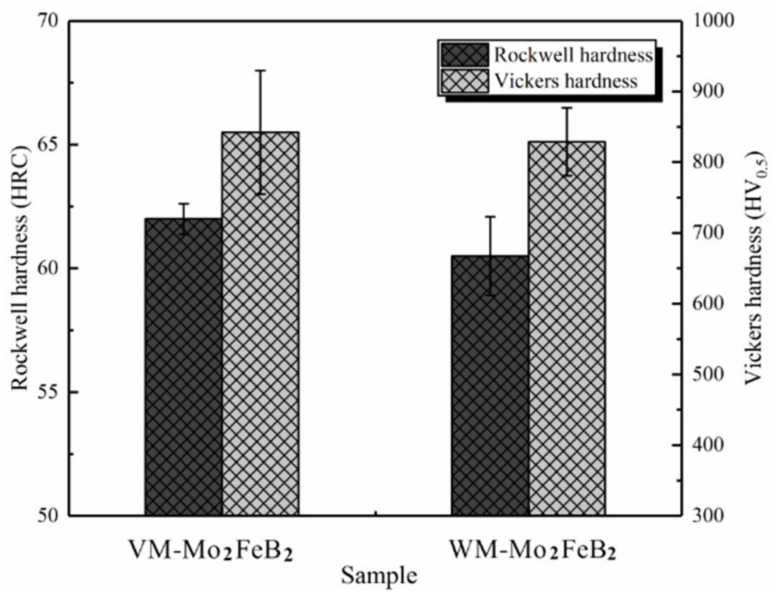
Rockwell hardness and Vickers hardness of VM-Mo_2_FeB_2_ and WM-Mo_2_FeB_2_.

**Figure 9 materials-14-00046-f009:**
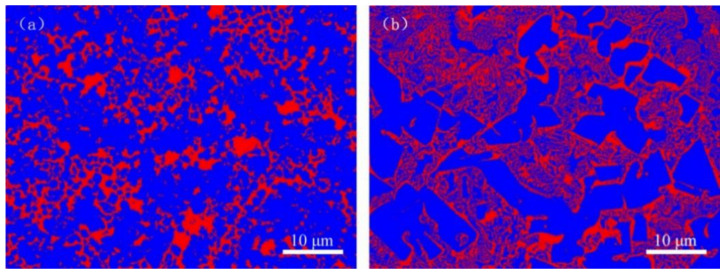
Images of calculation of hard phase fraction: (**a**) VM-Mo_2_FeB_2_, (**b**) WM-Mo_2_FeB_2_.

**Figure 10 materials-14-00046-f010:**
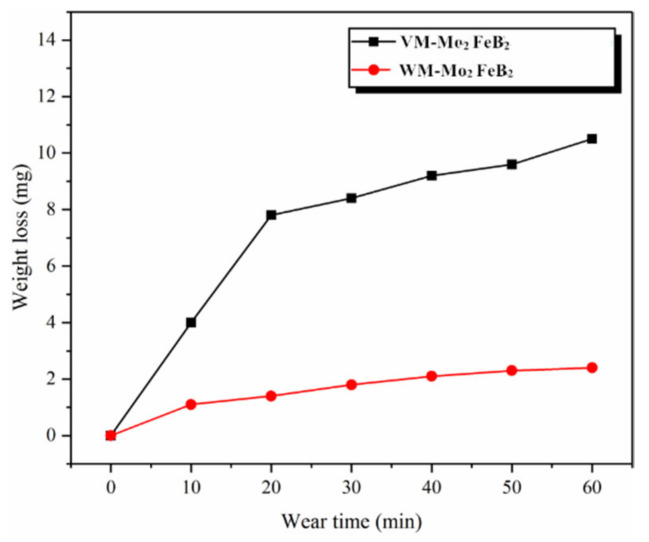
Wear weight loss of VM-Mo_2_FeB_2_ and WM-Mo_2_FeB_2_.

**Figure 11 materials-14-00046-f011:**
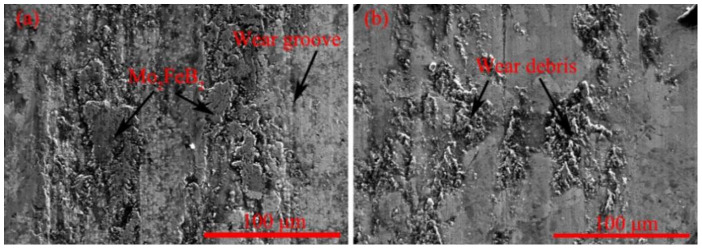
Wear morphology of different samples: (**a**) VM-Mo_2_FeB_2_, (**b**) WM-Mo_2_FeB_2_.

**Figure 12 materials-14-00046-f012:**
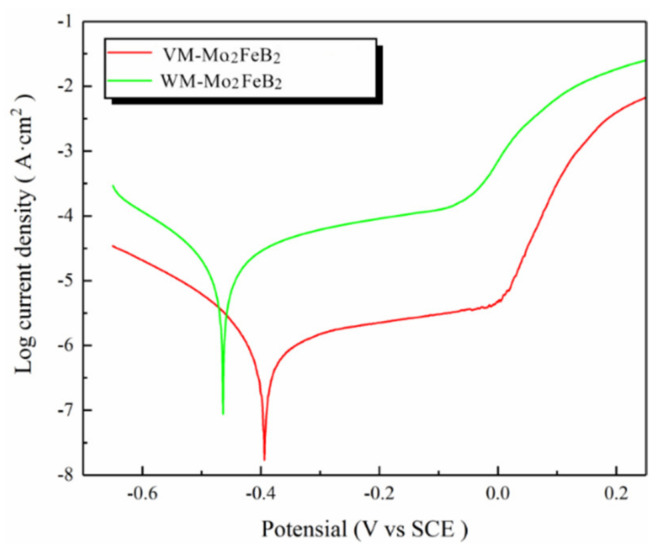
Potentiodynamic polarization curves of different samples in 3.5 wt% NaCl solution.

**Figure 13 materials-14-00046-f013:**
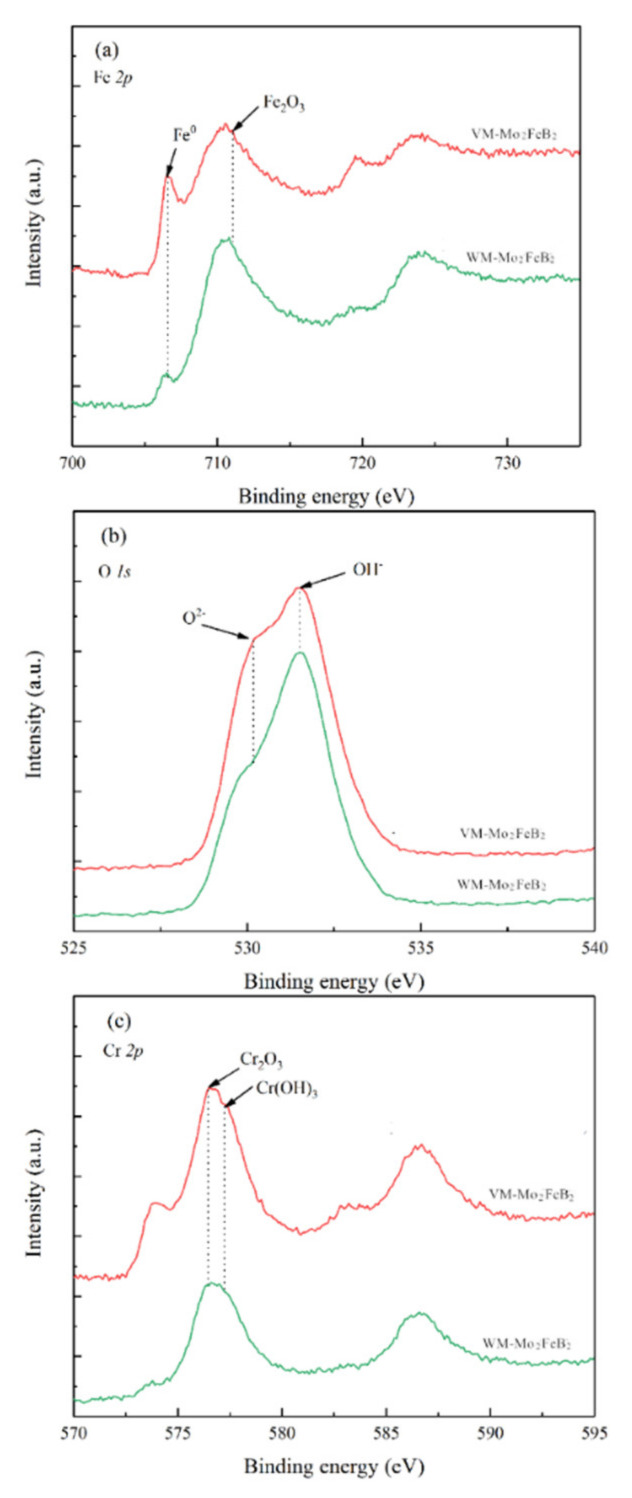
XPS spectra of Fe 2p, O 1s and Cr 2p for passive films formed on VM-Mo_2_FeB_2_ and WM-Mo_2_FeB_2_ in 3.5 wt% NaCl solution: (**a**) Fe 2p, (**b**) O 1s and (**c**) Cr 2p.

**Figure 14 materials-14-00046-f014:**
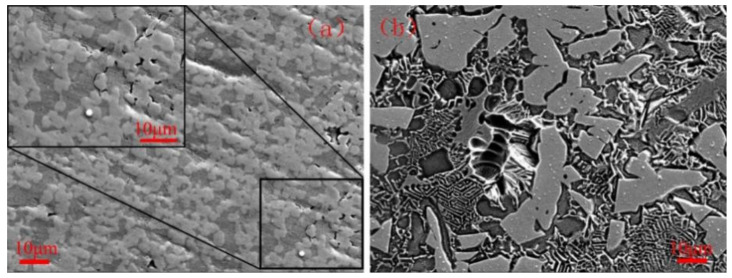
Corrosion morphologies of VM-Mo_2_FeB_2_ and WM-Mo_2_FeB_2_: (**a**) VM-Mo_2_FeB_2_ and (**b**) WM-Mo_2_FeB_2_.

**Table 1 materials-14-00046-t001:** The technical parameters of the alloy powders.

Powder	Mean Particle Size (μm)	Chemical Composition (wt%)
Fe	80	C < 0.1, N < 0.1, O < 0.2, Bal Fe
Ni	75	Ni ≥ 99.5, C < 0.15, Si < 0.01, Bal Ni
FeB	120	B = 22%, C < 0.27, Si < 0.71, Bal Fe
Mo	110	Fe < 0.002, O < 0.1, Si < 0.001, Bal Mo
Cr	75	O < 0.2, Fe < 0.18, N < 0.045, Bal Cr

**Table 2 materials-14-00046-t002:** Chemical composition of 304 stainless steel (wt%).

C	Mn	P	S	Si	Cr	Ni
0.05	1.2	0.015	0.01	0.2	18.5	9.2

**Table 3 materials-14-00046-t003:** Technological parameter of TIG (Argon tungsten arc welding).

Parameter	Value
Protective gas	99.9% pure argon
Tungsten diameter; Current	3.2 mm; 140 A
Voltage	18–20 V
Flow rate of argon	12 L/min
Speed of welding	100 mm/min

**Table 4 materials-14-00046-t004:** EPMA (Electron Microprobe) results of 1zone in VM-Mo_2_FeB_2_.

Element	Mass Fraction (%)	Atom Fraction (%)
B	11.008	44.6094
Cr	18.376	15.4812
Fe	21.890	17.1708
Ni	1.689	1.2599
Mo	47.038	21.4785
Total	100	100

**Table 5 materials-14-00046-t005:** Corrosion parameters of different samples evaluated by potentiodynamic polarization tests in 3.5 wt% NaCl solution.

Sample	*E*_corr_ (V)	*I*_corr_ (μA/cm^2^)	*E*_pit_ (V)
VM-Mo_2_FeB_2_	−0.38	1.14	0.02
WM-Mo_2_FeB_2_	−0.46	37.6	−0.04

*E*_corr_, *I*_corr_ and *E*_pit_ represent corrosion potential, corrosion current density and pitting potential, respectively.

## Data Availability

There are no restrictions on the availability of data in this study.

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
