# Peer review of "Comparison of Structure and Properties of Mo2FeB2-Based Cermets Prepared by Welding Metallurgy and Vacuum Sintering"

_materials, 2020, doi:10.3390/ma14010046_

Round 1

Reviewer 1 Report

The authors describe their work on preparing ternary metal borides through welding with the example of Mo2FeB2 and focus on its preparation and properties. This work is an interesting addition to the work being done on metal boride ceramics, however, the authors do not do a good job of presenting and framing the rationale and background for their work. The following points must be addressed in order to consider the revised manuscript for publication:

  1. The authors should provide more background information on the recent work being done in ternary metal borides. In the past few years, several novel efforts have contributed much knowledge to the structure and preparation of crystalline ternary metal borides. Considering the strong bonding existing within these materials that contributes to their incredible refractory properties, it is crucial to reference all efforts where metal ions are partially replaced in the metal boride crystal structure. Please expand the introduction section to detail current progress in metal boride chemistry. The following significant works are missing from the introduction section:
  • Ramachandran, R. and Salguero, T.T., 2018. Nanostructuring of Strontium Hexaboride via Lithiation. Inorganic chemistry57(1), pp.4-7.
  • Cahill, J.T., Alberga, M., Bahena, J., Pisano, C., Borja-Urby, R., Vasquez, V.R., Edwards, D., Misture, S.T. and Graeve, O.A., 2017. Phase stability of mixed-cation alkaline-earth hexaborides. Crystal Growth & Design17(6), pp.3450-3461.
  • Verger, L., Kota, S., Roussel, H., Ouisse, T. and Barsoum, M.W., 2018. Anisotropic thermal expansions of select layered ternary transition metal borides: MoAlB, Cr2AlB2, Mn2AlB2, and Fe2AlB2. Journal of Applied Physics124(20), p.205108.
  1. In the Introduction section, the authors speculate on the rationale for using their chosen methodology without backing it with prior literature. Please include relevant references when making claims about the benefits of the welding metallurgical method.

  1. Please include ICDD/JCPDS numbers for your XRD reference patterns (either in figure legend or in the text).

  1. Equations 1 and 2 are not balanced chemical equations. I understand that it is hard to balance them as the ratios of the metal will vary. So please refer to the formula as x, (1-x) etc. when writing the reactions. Furthermore, “Liquid” is not informative in the equation. Please elaborate on what that means.

  1. In Figure 3, please depict the lattice planes and X,Y,Z axes.

  1. In the results and discussion section, please include references when making speculator statements like “At the same time, the (100) direction of Mo2FeB2 crystal, that is, the c-axis direction, is the preferred direction for grain growth.”, “Some reasons can explain this phenomenon:….” etc.

Author Response

Dear reviewer:

We gratefully appreciate for your valuable comment.According with your advice,we amended the relevant part in manuscript.Some of your questions are answered below.

1) We are very sorry for the lack of research results of borides in recent years in the preface. We have also read the research contents of some scholars, especially the literature recommended by you, and we feel greatly benefited. In the foreword, we add the research results in recent years, such as the nanostructure of strontium hexaborate, the anisotropic thermal expansion properties of layered ternary transition metal borides MoAlB,Cr2AlB2,Mn2AlB2,Fe2AlB2.

2) As for the lack of literature support for the advantages of welding methods in your preface, we have found some practical applications of welding metallurgy by consulting the data. We think this can explain the advantages of welding from some aspects.

3) Sorry for the lack of the ICDD/JCPDS numbers, which is our negligence, we have added the ICDD/JCPDS numbers to the annotation in Figure 1.

4) Equations 1 and 2 are not equilibrium chemical equations, because (Mo, Fe, Cr) 3B2 represents that Mo and Fe atoms in the lattice of Mo2FeB2 will be replaced by Cr to form a solid solution, so it is difficult to balance with your method. For the information of liquid phase, we think it is eutectic liquid phase of Fe-B through the ternary phase diagram of Mo-Fe-B.

5) In Figure 3 and the notes below, we show the lattice details.

6) In the literature of Wang Hongquan scholars, we found examples.

Once again,thank you very much for your comments and suggestions.

Reviewer 2 Report

Dear Authors,

The article entitled "Comparison of structure and properties of Mo2FeB2-based cermets prepared by welding metallurgy and vacuum sintering" makes a structural and property comparison of Mo2FeB2-based cermets. The work is well developed, the characterization is complete and the results are well discussed. I consider that the work developed is interesting for the field and allows to evaluate two processes of obtaining pieces depending on the assessment. This is useful even when the material shows slightly different properties and structure. Based on this, I recommend the acceptance of this work for publication. I only have a small comment, the deconvolution of the XPS signals should be done and shown in figure 13 since this is very useful to evaluate the presence of other bonding energies or oxidation states.

Best regards

Author Response

Dear reviewer:

Thank you very much for your recognition of our work, which will be of great significance to us. The deconvolution of the XPS signal you said should be completed and displayed in Figure 13. Due to our negligence in the design of the experiment and the limitation of the real equipment, we are sorry that we can't add it for the time being. We will certainly pay attention to the completion of this aspect in our subsequent work, and hope to get your understanding.

Once again,thank you very much for your comments and suggestions.

Reviewer 3 Report

The paper entitled ”Comparison of structure and properties of Mo2FeB2-based cermets prepared by welding metallurgy and vacuum sintering” presents some interesting experimental research. The authors performed a lot microstructural investigation.

In order to be published, please add some information in introduction regarding the presence of impurities in the materials and how they affect the structure and corrosion resistance.

Please correct the conclusion label. ”44”

Author Response

Dear reviewer:

We gratefully appreciate for your valuable comment.We have revised the content of the introduction and added some necessary elements to your question. We hope we can get your approval. We are sorry for some details in the article, and have made corrections.

Once again,thank you very much for your comments and suggestions.

Reviewer 4 Report

Title: Comparison of structure and properties of Mo2FeB2-based cermets prepared by welding metallurgy and vacuum sintering

The manuscript can be accepted for publication after minor revision. Some comments are given below:

  • The research question and novelty of this work is missing in introduction part. Also, the introduction part needs improvement. I recommend the authors to rewrite introduction part and specify the novelty of this work in introduction section.
  • Figure 1 should be indexed with crystalline peaks and planes.
  • Chemical formula of PVB should be given in materials and methods.
  • The conclusion part should be in bullets format.
  • There are some typos. I recommend the authors to deeply read and study the entire manuscript and remove the typing, spelling and grammatical errors.
  • The English language should be proofread.

Decision: Minor Revision

Author Response

Dear reviewer:

It's a great honor that this article has been reviewed by you. Thank you very much for your comments. For the problems in the introduction part, we re add some achievements about boride research, and introduce the practicability and novelty of this paper through engineering examples.Sorry for the lack of the ICDD/JCPDS numbers, which is our negligence, we have added the ICDD/JCPDS numbers to the annotation in Figure 1.We have also adjusted the format of the conclusion. We are sorry for the typographical, spelling and grammatical errors in the text. We have reviewed the whole article and carefully corrected the typing, spelling and grammar errors of the article. We hope to get your approval.

Once again,thank you very much for your comments and suggestions.